# Transfer RNA Mutation Associated with Type 2 Diabetes Mellitus

**DOI:** 10.3390/biology12060871

**Published:** 2023-06-16

**Authors:** Fanny Rizki Rahmadanthi, Iman Permana Maksum

**Affiliations:** Departement of Chemistry, Faculty of Mathematics and Natural Sciences, Universitas Padjadjaran, Sumedang 45363, Indonesia; fanny19002@mail.unpad.ac.id

**Keywords:** tRNA genes, mutation, diabetes mellitus

## Abstract

**Simple Summary:**

Diabetes has a high mortality rate. Diabetes mellitus is a state of hyperglycemia or high glucose levels caused by the inability of the body to produce insulin, insulin resistance, or both. One of the causes of diabetes is the occurrence of mutations in mitochondrial genome genes and the loss of transfer RNA modification. Transfer RNA genes are part of the mitochondrial DNA genome that act as adapters and are key for protein synthesis. In this review, we discuss the structure of transfer RNA, mutations associated with and their relation to various diseases, as well as mutations associated with type 2 diabetes mellitus. In addition, methods that have been used to identify mutations and ideas for treatment are discussed.

**Abstract:**

Transfer RNA (tRNA) genes in the mitochondrial DNA genome play an important role in protein synthesis. The 22 tRNA genes carry the amino acid that corresponds to that codon but changes in the genetic code often occur such as gene mutations that impact the formation of adenosine triphosphate (ATP). Insulin secretion does not occur because the mitochondria cannot work optimally. tRNA mutation may also be caused by insulin resistance. In addition, the loss of tRNA modification can cause pancreatic β cell dysfunction. Therefore, both can be indirectly associated with diabetes mellitus because diabetes mellitus, especially type 2, is caused by insulin resistance and the body cannot produce insulin. In this review, we will discuss tRNA in detail, several diseases related to tRNA mutations, how tRNA mutations can lead to type 2 diabetes mellitus, and one example of a point mutation that occurs in tRNA.

## 1. Introduction

Mitochondria are cellular organelles that produce energy in the form of adenosine triphosphate (ATP) through the process of oxidative phosphorylation (OXPHOS); thus, they are referred to as the cellular power plant. Each cell contains hundreds to thousands of copies of the mitochondrial DNA genome. The human mitochondrial DNA genome encodes a total of 37 genes, of which 13 are used for the respiratory complex, 22 transfer RNAs (mt-tRNA), and 2 ribosomal RNAs (mt-rRNA). The mitochondrial protein synthesis machine, mt-tRNA, acts with the aminoacyl-tRNA synthetases, elongation factors, and ribosomes to translate the thirteen genes [1,2]. tRNA plays a crucial role in protein synthesis by serving as a key component in the translation mechanism. This synthesis is a highly complex process requiring many components [3] and the accuracy is typically maintained by the standard codons and anticodons of the tRNA, including wobble positions [4].

Human mitochondrial tRNA has received attention due to the correlation between point mutations in tRNA genes and various neuromuscular and neurodegenerative disorders [5]. However, in other cases, mutations occurring in these tRNA genes can damage the respiratory chain, leading to mitochondrial dysfunction, which significantly contributes to the development of diabetes mellitus. Initially, tRNA mutations were reported in individuals with mitochondrial encephalomyopathy, lactic acidosis, and stroke-like episodes (MELAS) disease, specifically associated with the tRNA^leu^ gene [6]. Furthermore, tRNA mutations have been linked to myoclonic epilepsy with ragged red fibers (MERRF) disease [7]. Since then, numerous studies have continued to establish connections between tRNA mutations and various other diseases including diabetes mellitus. Moreover, point mutations in the tRNA gene can lead to diabetes mellitus syndromes such as maternally inherited diabetes and deafness (MIDD), MELAS, MERRF, and polycystic ovary syndrome (PCOS) with diabetes mellitus [8]. According to the International Diabetes Federation (2021), around 537 million adults globally (20–79 years) have been diagnosed with diabetes with 6.7 million reported deaths due to diabetes. Furthermore, 44.7% or about 239.7 million adults living with diabetes are unaware of their status; thus, early diagnosis is important to prevent or delay complications and avoid premature death [9].

Based on data reported on MITOMAP (2023) as a human mitochondrial genome database [10], tRNA mutations are the most common mutations compared to rRNA, with 377 mutations. All tRNA genes are reported to have at least one mutation and the mutations can be identified using relatively inexpensive sequencing technology. This review provides a detailed explanation of tRNA genes and the relationship between tRNA mutations and diseases, particularly type 2 diabetes mellitus. We will also present an example of a specific tRNA mutation and discuss relevant in silico and in vitro research.

## 2. Search Strategy

Google and Google Scholar were searched until the end of February 2023 to gather information from general, in vitro, and bioinformatics studies (in silico). If possible, some terms were restricted to title only, i.e., transfer RNA (tRNA) mutation and the relationship of tRNA mutation with diabetes mellitus. The results were sorted by relevance for screening. The authors referred to the MITOMAP database (http://www.mitomap.org/MITOMAP, accessed on 25 February 2023) for the reported mutations.

## 3. Structure and Function of Transfer RNA Genes

The mitochondrial DNA genome in humans is circular and consists of 16,569 bp. The mitochondrial genome encodes 13 protein subunits which serve as the core respiratory chain (7 subunits of complex I: ND1, ND2, ND3, ND4, ND4L, ND5, ND6; 1 subunit of complex III: cytochrome b; 3 subunits of complex IV: cytochrome c oxidase 1 cytochrome c oxidase 2, cytochrome c oxidase; and 2 subunits of complex V: ATP6 and ATP8), 2 ribosomal RNAs (rRNAs), and 22 mitochondrial transfer RNAs (tRNAs).

tRNAs are relatively small and are formed from single-strand RNA (ssRNA) folded into a 3D structure. tRNAs in bacteria and eukaryotes are between 73 and 93 nucleotides long with a molecular weight of about 24,000–31,000. Most tRNAs have a guanylate residue (pG) at the 59 end and all have the trinucleotide sequence CCA at the 39 end. When drawn in 2D, the hydrogen bond patterns of all tRNAs form a four-leaf clover structure; the longer tRNA has a short fifth arm or extra arm and the L-shaped tRNA is bent in 3D [11]. Generally, the cloverleaf structure consists of an acceptor arm, D-loop, anticodon stem, variable region, and TψC loop [12]. The tRNA arm serves as a carrier for specific amino acids which are esterified by the carboxyl group to the 2′- or 3′ hydroxy group of residue A at the 3′ end of the tRNA. Each anticodon loop contains three specific base anticodons for the mRNA codon to produce the appropriate amino acid during translation, while each stem will undergo maturation and be filled with the appropriate amino acid [13,14]. In addition, the D-loop is composed of the unusual nucleotide dihydrouridine (D), and the TψC loop is composed of ribothymidine (T), and pseudouridine (ψ) which has a carbon–carbon bond between the base and ribose. The D-loop and TψC loops play a role in the tRNA folding and the TψC loop interacts directly with rRNA subunits.

tRNA acts as an adapter molecule that facilitates the conversion of the genetic code into amino acid sequences, thus playing a role in protein synthesis, namely in the activation of amino acids. Each amino acid will attach covalently to a specific tRNA with the help of an enzyme, aminoacyl-tRNA-synthetase. The end of the tRNA will pair with the appropriate amino acid and the other end will pair with the anti-codon on the messenger RNA (mRNA) [15,16]. When the tRNA is attached in a process known as aminoacylation, the tRNA is said to be “charged”. This aminoacylation occurs in the cytosol [11].

tRNA biogenesis or maturation involves synthesizing the initial transcript, removing residues at the 59 and 39 ends, adding CCAs, splicing introns (if present), and modifying nucleotide residues. The primary transcript of the tRNA gene contains additional 5′ and 3′ sequences which are then removed by a set of nucleases before the tRNA introns are spliced by endonucleases and the resulting fragments are joined by RNA ligase. Then, the CCA sequence at the 3′-terminus is added post-transcriptionally by a CCA-enhancing enzyme [17,18].

After transcription, the tRNA will undergo post-transcriptional processing such as modification of sugars by various enzymes. These chemical modifications serve several purposes, such as enhancing the stability of the tRNA structure, enabling proper interactions with other molecules, and protecting the tRNA from degradation. There are 43 types of stable tRNA modifications in humans, with each chemical structure counted as one modification [19]. To maintain normal function, the U34 position is required for modification of two related taurines in tRNAs, such as τm^5^U for tRNA^Leu(UUR)^ and tRNA^Trp^ and τm^5^s^2^U for tRNA^Glu^, tRNA^Lys^, and tRNA^Gln^. Another important chemical modification is at position 37, specifically in the anticodon stem sequence. Modifications to this position preserve the function of the A-site anticodon and maintain an accurate translational reading frame [20,21,22].

The tRNA gene represents only a small position of the entire human mitochondrial genome [23] which is divided into two strands with each strand containing 22 types of tRNA, namely glutamic acid, alanine, asparagine, cysteine, tyrosine, serine, glutamine, and proline occur at the L-strand whereas phenylalanine, valine, leucine, isoleucine, methionine, serine, tryptophan, aspartic acid, lysine, glycine, arginine, histidine, and threonine occur at H-strand (Table 1). Protein synthesis begins with the start codon, AUG, and ends with one of three stop codons such as UGA, UAG, and UAA. Usually, 61 codons encode 20 different amino acids [24].

## 4. Some Diseases Associated with Mitochondrial Transfer RNA Mutations

Translation errors can occur due to various mechanisms. A single nucleotide change commonly known as a substitution mutation at a codon that codes for an amino acid can impact the function, efficiency, and stability during protein synthesis. Additionally, mutations can cause the transfer RNA (tRNA) structure to become unstable and susceptible to degradation. Consequently, the aminoacylation becomes inefficient since the enzyme involved in this process specifically recognizes tRNA [25]. There are two possibilities; it could be that the mutation causes a loss of function in the tRNA or that the function has increased due to a mutation [24]. Currently, the most plausible explanation for the occurrence of tRNA mutations is the absence of a robust DNA repair system in mitochondrial DNA. Unlike nuclear DNA, mitochondrial DNA lacks efficient mechanisms for repairing DNA damage, making it more susceptible to mutations.

Some mutations in mitochondrial tRNA reported in MITOMAP have been associated with several (Table 2) and are plotted on the human mitochondrial genome in Figure 1.

Based on Table 2 and Figure 1, it is evident that a single mutation can be associated with multiple diseases. Conversely, a specific disease is not exclusively linked to only one point mutation. It is important to note that ongoing research is constantly uncovering new mutations, and the provided table may not encompass all reported mutations. However, it can be inferred that each tRNA has the potential to undergo mutations.

## 5. Association between Transfer RNA Mutations and Type 2 Diabetes Mellitus

Transfer RNA (tRNA) mutations are associated with various diseases but this review will focus on the association with diabetes mellitus (Figure 2). Diabetes mellitus can be considered the “mother of disease” because it is associated with various medical conditions, such as hypertension, heart disease, stroke, and deafness [82].

Diabetes mellitus is a metabolic disorder characterized by high blood sugar levels or hyperglycemia, which is caused by abnormalities in insulin secretion, insulin resistance, or both. Diabetes mellitus is classified into two types, namely, type 1 diabetes mellitus (T1DM) or known as called insulin-dependent diabetes mellitus (IDDM), and type 2 diabetes mellitus (T2DM) or non-insulin-dependent diabetes mellitus (NIDDM). T1DM is the result of the destruction of pancreatic β cells which causes insulin deficiency, whereas T2DM is caused by reduced insulin secretion by pancreatic β cells and insulin resistance [83,84,85,86]. Insulin deficiency can occur via damage to pancreatic B cells desensitization or decreased function of glucose receptors in the pancreas, and damage to insulin receptors in peripheral tissues [87]. T1DM accounts for 5–10% of cases, with most diabetes cases being T2DM [88,89,90]. Multiple factors contribute to the risk of developing T2DM, such as obesity, genetic factors (heredity), and metabolism which will interact with each other and cause disturbances in insulin secretion and mechanism of action [91]. Figure 2 shows the link between hyperglycemia as an early stage of diabetes and heredity that cause T2DM, one of which is a tRNA mutation in mitochondrial DNA.

The American Diabetes Association (ADA) classifies “mitochondrial diabetes” under the category of “Other, genetic defect of the β cell”, which can be caused by mutations in mitochondrial DNA [83]. Generally, this type of diabetes occurs in adults under the age of 70 [92]. In addition, T2DM caused by genetic factors also generally involves mutations in mitochondrial DNA (Figure 2). This mutation is usually inherited maternally because more mitochondria are found in egg cells and after fertilization, the mitochondria in spermatozoa will die and leave the mitochondria of the egg [93].

The relationship between tRNA gene mutations and diabetes mellitus starts from high blood sugar levels in the body or when the body is experiencing hyperglycemia, thereby triggering insulin secretion. Glucose is transported into the pancreatic β cells by the glucose transporter and undergoes rapid phosphorylation into glucose 6-phosphate via glycolysis with the help of the precursor enzyme, glucokinase. The product of glycolysis is pyruvate which enters the mitochondria with the help of pyruvate carboxylase. The mitochondria have five complexes (I, II, III, IV, and V) and the pyruvate is converted to acetyl co-a through the citric acid cycle in complex II. Then, glucose is converted to ATP as energy in complex V which acts as a signaling molecule for insulin secretion because cells have K^+^ channels that are sensitive to ATP, helping the cell to keep K^+^ channels closed causing membrane depolarization. The Ca^2+^ channel will open to allow Ca^2+^ to enter the cell and trigger the insulin granules to undergo exocytosis to release insulin for secretion [94,95,96,97]. However, when there is a mutation in the mitochondrial gene (in the protein subunit, tRNAs, or rRNAs), ATP production is impaired, thus the K^+^ channels fail to close, the membrane is not depolarized, and the Ca^2+^ channels do not open. Consequently, insulin secretion does not occur, and cellular glucose levels remain elevated.

Another mechanism related to point mutations in the tRNA of the mitochondrial genome is that diabetes is dependent on insulin resistance (IR). IR is the inability of the body to detect the presence of insulin, so it cannot take glucose from the blood. IR can interfere with the insulin signaling pathway. When a mutation occurs in the mitochondrial genome, reactive oxygen species (ROS) (as a by-product of mitochondria) can increase insulin sensitivity but high levels of ROS in an oxidative environment can alter mitochondrial function and lead to the development of IR, dysfunction of pancreatic β cell, as well as glucose tolerance [98,99,100].

The relationship between mutations that occur in mitochondrial DNA and respiratory function can be explained using an in silico approach. Mutations do not only occur in tRNA genes but can also in protein subunits, so although protein synthesis can still occur, mutations in these proteins can disrupt ATP production. Subunit proteins within the respiratory complex play essential roles in proton translocation and transfer, and mutations affecting these proteins can impede ATP production. Mutations within the coding region directly impact the function of protein subunits, whereas mutations in tRNA are associated with premature proteins.

Intriguingly, several mutations in subunit proteins have been identified in diabetes mellitus. For instance, Maksum et al. (2017) [101] conducted in silico analysis of the G9053A mutation which occurs in the ATP6 gene and is found in individuals with diabetes mellitus and cataracts, revealing that this gene is part of complex V directly affecting ATP synthesis. In addition, in silico studies of the T10609C and C10676G mutations by Destiarani et al. [102] showed that these mutations occurred in the ND4L subunit and affect proton translocation because they occur in complex I of the respiratory chain. Interestingly, this mutation is also found in patients with cataracts.

Zhou et al. [8] proposed a scheme highlighting tRNA dysregulation in diabetes mellitus. The scheme begins with tRNA transcription by RNA polymerase III, which is susceptible to mutations, leading to the production of mutant tRNAs which can impair tRNA aminoacylation and tRNA modification, thereby rendering the tRNA defective. Additionally, both pre-tRNA and mature tRNA can experience stress, resulting in the formation of tRNA derivatives. Furthermore, tRNA mutations can also give rise to tRNA derivatives and modified pre-tRNA and mature tRNA, which undergoes post-transcriptional modifications leading to tRNA defects due to deficiencies in the enzymes involved in these modifications. Post-transcriptionally modified tRNAs that undergo aminoacylation become tRNAs that participate in protein translation may experience abnormal tRNAs due to the presence of lipotoxicity factors.

tRNAs also often experience loss of modifications, commonly called “tRNA modopathies”. More than 40 types of human tRNA modopathies play an important role in protein synthesis as they are responsible for regulating the structure and stability of tRNAs and decoding the genetic information in messenger RNA (mRNAs) [98]. When tRNA undergoes modopathies, it causes the structure of the tRNA to change from wild-type to mutant so that it has an impact on translation so that the mRNA that is translated due to the wrong process will be a protein whose function also changes. When the function of a protein changes, it affects the subunit function complex and leads to aberrant insulin production, contributing to diabetes (Figure 2). Cdk5 regulator associated with protein 1-like 1 (CDKAL1) is a gene identified as the binding protein to the activator of cyclin-dependent kinase (CDK5), a previously uncharacterized gene, both of which are associated with a risk of T2DM. CDKAL1-mediated 2-methylthio modification of tRNA^Lys(UUU)^ affects the stability of codon–anticodon interactions and contributes to translation. If there is a lack of CDKAL1 there will be errors in proinsulin translation (mistranslation), as well as the downregulation of metallothionein. There both impact unfolded proteins and collectively, can inhibit pancreatic β cell function and lead to the development of T2DM [103].

The A3243G mutation in the tRNA gene has been found in patients with diabetes mellitus with deafness or maternally inherited diabetes and deafness (MIDD), which is classified as a causal mutation [104]. This mutation is the most common pathogenic mutation in mitochondrial DNA, with a prevalence ranging from 0.95 to 16.3 per 100,000 individuals [105,106]. This mutation follows the mechanism previously mentioned, interfering with the function of tRNA to properly assemble proteins; therefore, there is insufficient ATP to open K^+^ channels, so cells cannot secrete insulin in response to hyperglycemia which leads to T2DM [107].

The A3243G mutation occurs in the tRNA^Leu(UUR)^ gene, which is responsible for encoding the UUR codon (R = A or G). This mutation occurs at the mtDNA binding site for the transcription factor (mTERF), leading to a decrease in mTERF affinity [14,108]. The A3243G mutation refers to a change from adenine to guanine at position 3243 in the mitochondrial genome or a mutation occurring at position 14 within the tRNA structure.

Extensive research has been conducted to understand the pathogenesis of the A3243G mutation. It has been found to affect various aspects of the physiological state of the tRNA^Leu(UUR)^ gene, including structural stability, aminoacylation rate, gene codon recognition, and methylation (post-transcriptional modification) [109]. Specifically, the A3243G mutation occurs at position A14 in the tRNA structure, where A14 is hydrogen-bonded with U8, thereby disrupting the A-U bond and weakening the arm structure of the tRNA, leading to increased openness and potential dimerization with other mutant forms. The mutation induces the formation of a dimer complex through the introduction of a palindromic hexanucleotide sequence, 5-GGGCCC-3, in the D-loop. In silico studies have shown that the molecular weight of the tRNA structure doubles when the A3243G mutation occurs, indicating the formation of a dimer structure [110].

The formation of dimers and disruption of the U8:A14 base pair in the A3243G mutant significantly contributes to a decreased rate of aminoacylation. Aminoacylation is facilitated by the enzyme aminoacyl-tRNA synthetase, which catalyzes the ester reaction between the OH group on the tRNA and the carboxylate of the corresponding amino acid. Wittenhagen and Kalley found that the A3243G mutation inhibited aminoacylation of the native tRNA^Leu(UUR)^ gene five times more than its mutant counterpart (tRNA mutant A3243G) [110].

In addition, research has been carried out on the rate of aminoacylation of five tRNA mutation variants, namely A3243G, A3252G, C3256T, T3271C, and T3291C. Tm measurements indicated that the structure of the A3243G and T3291C mutants was much more fragile than that of the other variants. Hence, deficient aminoacylation appears to be related to the structural instability of the tRNA [111]. In addition, molecular dynamics simulations on the dimeric tRNA^Leu^ with the A3243G heteroplasmy mutation in human mitochondria showed that the structure of the mutant tRNA^Leu^ dimer is more stable than the wildtype tRNA dimer based on the conformational energy and RMSD value. The value of mutant tRNA^Leu^ dimers is lower than the wildtype tRNA^Leu^ dimers as evidenced by the presence of more intermolecular hydrogen bonds [112]. Enzymes that assist aminoacylation will experience a decrease in their activity against dimeric substrates, causing the tRNA to become “uncharged” which means it does not carry amino acids so that it impacts the mitochondrial subunit protein which requires the amino acid leucine during translation. Recent research related to the dimer structure of the tRNA^Leu^ mutant by Puspita et al. (2023) confirmed that the dimer form is more stable and results in a decrease in the aminoacylation rate [113].

The A3243G mutation also disrupts the tertiary interaction at U8-A14-A21, leading to a disruption in uridine modification. One specific modification of uridine is the attachment of a methyltaurino group (taurine) at the C5 position, known as a taurino modification. This modification occurs at position U34 and is essential for recognizing UUG and UUA codons. The A3243G mutation results in a decrease in uridine modification [114].

Modification of uridine will affect the interaction between the AUU anticodon and UUG codon, causing the interaction between U at the anticodon and G at the codon to be stronger compared to unmodified interactions. The lack of uridine modification in the wobble position in MELAS patients reduces UUG codon translation but will not affect UUA codon translation. The reduced translation of UUG codons impacts genes rich in Leu(UUG) and contributes to the deficiency of complex I of the respiratory chain [114].

Furthermore, the A3243G mutation can also disrupt methylation at the G10 nucleotide position, which plays a crucial role in the recognition of methyltransferases. Methylation at this position is important as the absence of the CH_3_ group can affect the binding of tRNA with other elements, leading to defects in the function of mitochondrial coding enzymes such as leucyl-tRNA synthase and elongation factor protein [115,116].

Various methods have been developed to detect the presence of the A3243G mutation or other mutations including high-performance liquid chromatography (HPLC), dot-blot hybridization, pyrosequencing, two-dimensional electrophoresis, peptide nucleic acid, electrochemical biosensors, radiolabeled polymerase chain reaction (PCR), ligation-mediated PCR (LM-PCR), allele-specific PCR, PCR amplification of specific alleles (PASA), PCR-restriction fragment length polymorphism (PCR-RFLP), and quantitative PCR (qPCR) [28,36,117,118,119,120,121,122,123,124,125]. Each of these methods has its advantages and limitations with some being easy to use but may be less sensitive, whereas others may involve the use of radioactive materials, which can be a concern. Some methods may also be highly sensitive but time-consuming for detection purposes.

PCR is the gold standard method for detecting mutations and has several advantages such as being fast, sensitive, and accurate. Various PCR methods have also been used and are considered successful in identifying the A3243G point mutation, such as the quantitative allele-specific-PCR method which was used to demonstrate the accumulation of mutations with age in several tissues the number of A3243G mutations in mature tissue was up to 10 times higher that of infant tissue. This illustrates the progressive nature of point mutation accumulation in mtDNA during aging [125]. The LM-PCR method was utilized to identify mutations in blood samples from 233 diabetes mellitus patients and 126 healthy patients as controls, showing that five patients carried a heteroplasmy percentage of >0.01% [122]. Additionally, PCR-RFLP, PASA, and PCR single-strand conformation polymorphism (PCR-SSCP) methods have been employed to identify and study maternal lineage offspring for up to three generations. In a study involving 101 blood samples, two of them tested positive, and maternal inheritance could be identified using PASA [28]. Sriwidodo et al. (2008) [126] identified 50 diabetics from a Jakarta hospital using the two-base mismatch PASA method and PCR-RFLP with restriction enzymes. Maksum et al. (2013) [127] also conducted research on making a positive control or mutant template for the detection of the A3243G mutation using the site-directed mutagenesis method, by changing the normal position of 3243 A to G and then amplifying it. The Taqman-MGB-based qPCR method has also been used to detect and quantify the extent of mutational heteroplasmy with urine sediment samples, blood samples, and hair follicles. The qPCR results were then confirmed by PCR-RFLP and pyrosequencing. The sensitivity was as low as 0.1% with a quantification accuracy of up to 4% [128].

The severity of the mutation can be supported by the presence of other mutations, both primary and secondary mutations. Research has established a relationship between these mutations and ATP levels. A study was conducted on three subjects with both A3243G and T14502C mutations, three subjects with only the A3243G primary mutation, and three healthy individuals, demonstrating that ATP levels were further decreased in the presence of secondary mutations [129]. This can occur due to the interference of mutations with ATP formation, as previously described.

Recently, a tool for detecting ATP levels has been developed using an aptamer-based chemical biosensor with screen-printed carbon electrode/gold nano-particles (SPCE/AuNP). After undergoing a series of tests, this aptasensor method has shown potential for sample analysis [130].

## 6. Conclusions

Transfer RNA (tRNA) plays a crucial role in protein synthesis, specifically in the assembly of proteins involved in the respiratory chain complex but mutations can occur in tRNA, leading to defects or premature protein synthesis. Mutated tRNA is often associated with various diseases as it can disrupt mitochondrial function and affect insulin secretion such as type 2 diabetes mellitus (T2DM), which can lead to many syndromes. Among the most prevalent tRNA mutations is the A3243G mutation in tRNA^Leu^, which has been extensively studied in pathological investigations. Various methods have been developed for mutation detection, including polymerase chain reaction (PCR), PCR restriction fragment length polymorphism (PCR-RFLP), and quantitative PCR (qPCR). These techniques enable researchers to identify and analyze tRNA mutations reliably and accurately. After identifying tRNA mutations, we can proceed to the next step to prevent or treat T2DM. To achieve the right treatment for T2DM, it is necessary to know the potential relationship between drug targets and tRNA biogenesis so that tRNA does not undergo mutations.

## Figures and Tables

**Figure 1 biology-12-00871-f001:**
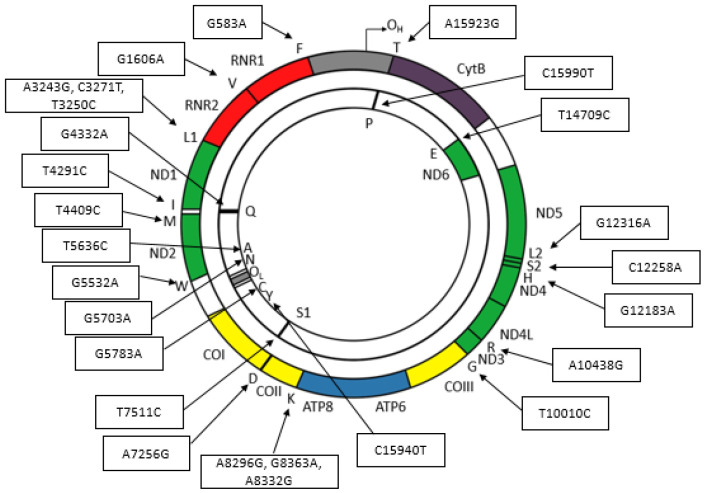
Mutation maps in human mitochondrial genome transfer RNA genes.

**Figure 2 biology-12-00871-f002:**
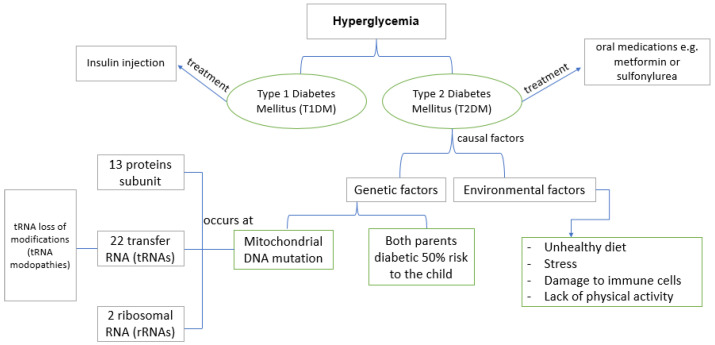
Schematic representation of the association between tRNA mutation and type 2 diabetes mellitus.

**Table 1 biology-12-00871-t001:** Size, location, and codon encoded by human mitochondrial transfer RNA genes. (non-polar, aliphatic; aromatic; polar, uncharged; positively charged, negatively charged).

Type of tRNA	Three-Letter Abbreviation	One-Letter Abbreviation	Gene	Codons	Size (bp)	Location in Genome
Phenylalanine	Phe	F	MT-TF	UUU UUC	71	577–647
Valine	Val	V	MT-TV	GUU GUC GUA GUG	69	1602–1670
Leucine (UUR)	Leu	L	MT-TL1	UUA UUG	75	3230–3304
Isoleucine	Ile	I	MT-TI	AUU AUC AUA	69	4263–4331
Glutamic Acid	Glu	E	MT-TE	GAA GAG	72	4329–4400
Methionine	Met	M	MT-TM	AUG	68	4402–4469
Tryptophan	Trp	W	MT-TW	UGG	68	5512–5579
Alanine	Ala	A	MT-TA	GCU GCC GCA GCG	69	5587–5655
Asparagine	Asn	N	MT-TN	AAU AAC	73	5657–5729
Cysteine	Cys	C	MT-TC	UGU UGC	66	5761–5826
Tyrosine	Tyr	Y	MT-TY	UAU UAC	66	5826–5891
Serine (UCN)	Ser	S	MT-TS1	UCU UCC UCA UCG	72	7445–7516
Aspartic Acid	Asp	D	MT-TD	GAU GAC	68	7518–7585
Lysine	Lys	K	MT-TK	AAA AAG	70	8295–8364
Glycine	Gly	G	MT-TG	GGU GGC GGA GGG	68	9991–10058
Arginine	Arg	R	MT-TR	AGA AGG	65	10405–10469
Histidine	His	H	MT-TH	CAU CAC	69	12138–12206
Serine (AGY)	Ser	S	MT-TS2	AGA AGG	59	12207–12265
Leucine (CUN)	Leu	L	MT-TL2	CUU CUC CUA CUG	71	12266–12366
Glutamine	Gln	Q	MT-TQ	CAA CAG	69	14674–14742
Threonine	Thr	T	MT-TT	ACU ACC ACA ACG	66	15888–15953
Proline	Pro	P	MT-TP	CCU CCC CCA CCG	69	15955–16023

**Table 2 biology-12-00871-t002:** Several transfer RNA mutations with associated diseases.

Syndrome	Point Mutation	tRNA Gene	Diseases	References
Diabetes mellitus (DM)	3243A>G	tRNA^Leu(UUR)^	MIDD ^1^, MELAS ^2^, PEO ^3^, Leigh syndrome, hearing loss	[26,27,28,29,30,31]
	8296A>G	tRNA^Lys^	Cardiomyopathy	[32,33]
	4291T>C	tRNA^Ile^	Myopathy, hypomagnesemia, and hypokalemia	[34,35]
	3271C>T	tRNA^Leu(UUR)^	DM	[36]
	12258C>A	tRNA^Ser(AGY)^	Hearing loss	[37,38]
Encephalomyopathy	1606G>A	tRNA^Val^	Hearing loss	[39,40]
	8363G>A	tRNA^Lys^	MERRF ^4^, autism, deafness	[41,42,43]
	8332A>G	tRNA^Lys^	Dystonia, MELAS, hearing loss	[44,45,46]
	583G>A	tRNA^Phe^	MELAS	[47,48,49]
	10010T>C	tRNA^Gly^	PEM ^5^	[50,51,52]
	10438A>G	tRNA^Arg^	Progressive encephalopathy	[53,54]
	7526A>G	tRNA^Asp^	MM	[55,56]
	4332G>A	tRNA^Gln^	MELAS	[57]
Mitochondrial myopathy (MM)	5703G>A	tRNA^Asn^	CPEO ^6^	[58,59]
	3250T>C	tRNA^Leu(UUR)^	MM, CPEO	[60,61]
	15990C>T	tRNA^Pro^	MM	[62]
	12316G>A	tRNA^Leu(CUN)^	CPEO	[63]
	4409T>C	tRNA^Met^	MM	[64,65]
	5532G>A	tRNA^Trp^	Gastrointestinal syndrome	[66]
	15923A>G	tRNA^Thr^	LIMM ^7^	[67,68,69]
	15940T>G	tRNA^Tyr^	Exercise intolerance	[70]
	5636T>C	tRNA^Ala^	PEO	[71]
Deafness	7511T>C	tRNA^Ser(UCN)^	SNHL ^8^	[72,73]
	5783G>A	tRNA^Cys^	Myopathy, SNHL	[74,75,76]
	12183G>A	tRNA^His^	SNHL, retinitis pigmentosa	[77]
	14709T>C	tRNA^Glu^	Mental retardation, cerebellar dysfunction, MIDD, MERRF	[48,78,79,80,81]

^1^ maternally inherited diabetes and deafness; ^2^ mitochondrial encephalomyopathy, lactic acidosis, and stroke-like episodes; ^3^ progressive external ophthalmoplegia; ^4^ myoclonic epilepsy with ragged red fibers; ^5^ protein-energy malnutrition; ^6^ chronic progressive external ophthalmoplegia; ^7^lethal infantile mitochondrial myopathy; ^8^ sensory neural hearing loss.

## Data Availability

Not applicable.

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
