# Peer review of "Transfer RNA Mutation Associated with Type 2 Diabetes Mellitus"

_biology, 2023, doi:10.3390/biology12060871_

Round 1
Reviewer 1 Report
General comments:
The manuscript entitled “Transfer RNA mutation associated with type 2 diabetes mellitus” by Fanny Rizki Rahmadanthi, and Iman Permana Maksum, is in the scope of the journal research topic and presents an interesting review article that addresses a carefully chosen topic with high interest for the biomedical community.
The present article reviews the tRNA genes and their mutation and connection with diseases, particularly type 2 diabetes mellitus.
Overall, this is an interesting and concise study. Nonetheless, some points should be amended:
Specific Comments:
1. Please check the following sentence, since it is confusing: “All the components needed for translation into mRNA are provided by the mitochondria, but the components for protein synthesis are provided by the cell nucleus and will then be transported to the mitochondria.”
2. My suggestion is to add a schematic representation of the Association of tRNA Mutation with Diabetes Mellitus Type 2.
3. My suggestion is also to add a Table summarizing the specific tRNA Mutation with specific aspects of Diabetes Mellitus Type 2 (described in subsection 4. Association tRNA Mutation with Diabetes Mellitus Type 2.).
4. The article should include a “Search strategy” section. It should be described when-which date, the literature search ended, and which were the main search terms and databases searched.
5. The literature search is adequate but should be updated. The majority of references cited in the paper are older than 5 years.
6. There are typos and errors in the aspect of English writing. Please check throughout the text to correct them.
7. All abbreviations need to be explained at first use in the text and Abstract and Figure legends. Some abbreviations are not introduced, and when once introduced the abbreviation should be exclusively used throughout the whole text.
6. There are typos and errors in the aspect of English writing. Please check throughout the text to correct them.
Author Response
Thank you very much for you comments. We have proofread the manuscript accordingly. Here the sertificate (input it in google drive) bit.ly/certificateproofread
Please see the attachment.

Reviewer 2 Report
See the attached file.

Author Response
Thank you for your comments and insights. Please see the attachment.

Reviewer 3 Report
The manuscript of “Transfer RNA mutation associated with type 2 diabetes mellitus” by Fanny Rizki Rahmadanthi and Iman Permana Maksum aims to review the role of tRNA mutations in mitochondrial DNA in the pathogenesis of type 2 diabetes mellitus and some other diseases. The authors have described in great detail only one type of these mutations, namely the A3243G mutation that occurs in the tRNALeu(UUR) gene, which is specifically responsible for encoding the UUR codon (R = A or G). It is known that the A3243G mutation causes markedly reduced ATP production, increased lactate production, impaired cell calcium homeostasis, increased ROS production, and reduced insulin secretion. The loss of these vital cellular functions can result in systemic disorders and the development of many diseases (PEO, Leigh syndrome, etc.). Besides, diabetes itself may be associated with various types of mutations, including mutations in genes encoding complexes of the mitochondrial respiratory chain. In this regard, the relationship between tRNA mutations in mtDNA and diabetes seems controversial. It is not clear why the authors chose this particular disease. In the conclusion section, there is not even a mention of diabetes mellitus. In addition, most of the text is devoted to long-known ideas about the structure of mtDNA, the etiology and pathophysiology of diabetes, some techniques enable researchers to identify and analyze tRNA mutations, etc.
The authors need to describe in more detail the relationship between point mutations in the transfer RNA (tRNA) of the mitochondrial genome and diabetes, taking into account recent data and review articles in 2023 (for example, DOI:10.3390/biom13010126; DOI: 10.3389/bjbs.2023.10884, etc.). In its current form, the manuscript does not significantly contribute to understanding the pathophysiology of diabetes caused by mtDNA mutations.
In addition, some sentences in the text are controversial. They don't sound quite scientific or need to be referred to research articles or paraphrased due to their distorted meaning. For example:
1 “The citric acid cycle is part of the mitochondrial complex II.”
2 “Moreover, one of the diseases frequently associated with point mutations in the tRNA gene is diabetes mellitus, which can manifest in the form of diabetes mellitus syndromes such as Maternally Inherited Diabetes and Deafness (MIDD), MELAS, MERRF, and Plycystic Ovary Syndrome (PCOS) with diabetes mellitus [8].”
3“Diabetes mellitus is often reffered to as the “mother of disease” because it can give rise to other medical conditions.”
4 “One type of diabetes that is categorized as DMT2, namely mitochondrial diabetes, one of which can be caused by mutations in mitochondrial DNA.” It should be noted that the American Diabetes Association (ADA) classifies “mitochondrial diabetes” under the category of “Other; genetic defects of the beta cell”.
Finally, the manuscript should be carefully revised by native English speakers or a professional language editing service to improve the readability and scientific sound.
The manuscript should be carefully revised by native English speakers or a professional language editing service to improve the readability and scientific sound.
Author Response
Thank you for your comments and insights.
Please see the attachment

Round 2
Reviewer 3 Report
1. The authors described in detail the prevalence and etiology of type 1 and type 2 diabetes mellitus, but there is no mention of the impact/prevalence of tRNA modifications to the pathogenesis of the disease.
2. Graphic scheme (Fig. 2) contains little information about current data on tRNA modopathies and are rather popular scientific drawings.
